# Susceptibility status of *Aedes aegypti* (Diptera: Culicidae) to public health insecticides in Southern Afar Region, Ethiopia

**Mohammed Seid**[1,2]*, **Esayas Aklilu**[1], **Abebe Animut**[1]

**1** Aklilu Lemma Institute of Pathobiology, Addis Ababa University, Addis Ababa, Ethiopia, **2** Department of Biology, College of Natural and Computational Sciences, Mattu University, Mattu, Ethiopia

* mohammed.seid.legas@gmail.com

**Data Availability Statement:** All relevant data are within the manuscript and its Supporting Information files.

**Funding:** The study was financially supported by Addis Ababa University, the Vice President for

## Abstract

Mosquito-borne viral diseases such as dengue fever, chikungunya, and yellow fever have been documented in Ethiopia since the 1960s. However, the efficacy of public health insecticides against *Aedes aegypti* that transmits these viruses remains poorly understood in the country, particularly in the Afar Region. Thus, the aim of the study was to assess the susceptibility status of *Ae. aegypti* to deltamethrin, permethrin, alpha-cypermethrin, pirimiphos-methyl, bendiocarb, and propoxur insecticides. Larvae and pupae of *Aedes* species were collected from Awash Arba, Awash Sebat, and Werer towns of the Afar Region of Ethiopia during July-October 2022, brought to the Aklilu Lemma Institute of Pathobiology, insectary and reared to adults. Non-blood-fed, 3–5 days-old females *Ae. aegypti* were exposed to pyrethroid, carbamate, and organophosphate insecticide impregnated papers in tube test following the standard guidelines. Knockdown rates were noted at 10 minutes interval until one hour. The mortality in mosquitoes was recorded 24 hours after 60 minutes of exposure. The mortality rates of *Ae. aegypti* exposed to propoxur were 87% in all the study towns. Similarly, 88% mortality in *Ae. aegypti* was recorded when tested with bendiocarb in Awash Sebat and Awash Arba towns. Suspected resistance of *Ae. aegypti* (95% mortality) to alpha-cypermethrin was observed in Awash Arba town. However, *Ae. aegypti* collected from all the three sites was observed to be susceptible to deltamethrin, permethrin, and pirimiphos-methyl. *Ae. aegypti* was resistant to 0.1% bendiocarb and 0.1% propoxur and possibly resistant to 0.05% alpha-cypermethrin. On the other hand, it was susceptible to 0.05% deltamethrin, 0.75% permethrin, and 0.25% pirimiphos-methyl. Thus, vector control products with deltamethrin, permethrin, and pirimiphos-methyl can be used in the control of adult *Ae. aegypti* in the Afar Region of Ethiopia. However, further studies should be carried out to evaluate the susceptibility status of *Ae. aegypti* to alpha-cypermethrin in the Awash Arba area.

## Introduction

The ever-increasing outbreaks of arboviral diseases have drawn public health attention in several countries around the world [1]. East African countries, including Kenya, Somalia,

Research and Technology Transfer Office through its thematic research grant. "The funder had no role in study design, data collection and analysis, the decision to publish, or the preparation of the manuscript.

**Competing interests:** The authors declared that there is no conflict of interest.

Djibouti, Eritrea, and Ethiopia are epicenters of outbreaks of arboviral diseases such as dengue, yellow fever and chikungunya [2–4]. In Ethiopia, yellow fever caused about 30,000 deaths from over 200,000 infections during 1960 and 1962 [5]. Recently, there have been yellow fever outbreaks in South Omo [6] and dengue fever outbreaks in Dire Dawa city administration, Somali Regional State, Gewane and Amibara districts of Afar Regional State [7, 8]. In addition, chikungunya outbreaks were reported in Dire Dawa city administration, Afar and Somali Regional States [9, 10]. All these outbreaks with potentially unrecognized and/or underreported deaths could not be associated with particular virus due to the highly limited laboratory facilities in the country.

Mosquitoes transmit over 90% of the viral pathogens through their bites, among which *Ae. aegypti* is the leading vector followed by *Ae. albopictus* [11, 12]. *Ae. aegypti* is anthropophilic and occurs in close proximity to human habitations [13, 14]. *Ae. aegypti* has two morphological subspecies (ecotypes); the domestic *Ae. aegypti aegypti* and sylvan *Ae. aegypti formosus* [15, 16]. In Ethiopia, *Ae. aegypti*, *Ae. bromeliae*, *Ae. vittatus*, *Ae. hirsutus*, *Ae. simpsoni* complex, and *Ae. africanus* have been implicated as potential vectors of viral diseases [6, 8, 17–19].

In the absence of effective therapeutic drugs and vaccines to treat and control mosquito-borne viral diseases, prevention largely depends on vector control with mainly public health insecticides [13]. Pyrethroids such as deltamethrin, alpha-cypermethrin, permethrin, and organophosphates pirimiphos-methyl and fenitrothion are among the public health insecticides used against *Aedes* mosquitoes [20]. However, the potential resistance to the insecticides may adversely affect the implementation of the control strategy in endemic areas [21]. Mosquitoes could resist insecticides through their inherent ability to detoxify and avoid contact either behaviorally or physiologically [22].

Evidence from many African countries indicates that *Anopheles* and *Aedes* mosquitoes developed resistance to insecticides [23–25]. For instance, in Ethiopia, malaria-transmitting *Anopheles arabiensis* has developed resistance to insecticides in several areas of the country, including the Amibara district of Afar Regional State [25, 26]. The cause of the resistance could be frequent use of the insecticides in indoor residual spraying and long-lasting insecticidal nets for the control of mosquito-borne diseases in the area [21] or in agriculture pest control [27]. However, the susceptibility status of *Aedes* mosquitoes to currently used insecticides is lacking in the country.

Recently, the Ethiopian Public Health Institute prepared an arboviral disease vectors surveillance and control guideline in 2021, emphasizing the use of chemical and environmental methods. Vector control by chemical methods such as fumigation, repellents, and residual insecticide surface treatments were implemented in Dire Dawa city administration and Somali Regional State during dengue fever and chikungunya outbreaks (EPH, 2021, unpublished). Unfortunately, there is a scarcity of evidence on the efficacy of the insecticides for the control of virus transmitting *Aedes* species in the Southern Afar Region in the presence of increased reports of viral disease incidence and prevalence. Therefore, assessing the effectiveness of the existing public health insecticides to control *Aedes* species is crucial to strengthening the arboviral disease prevention and control programs in the country. This inspired us to generate baseline data on the susceptibility status of *Ae. aegypti* to public health insecticides in the Southern part of the Afar Regional State of Ethiopia.

## Materials and methods

### Study sites

The study was carried out in Awash Arba, Werer, and Awash Sebat towns, Gabi-Rasu Zone (Zone 3), of Afar Regional State, Ethiopia. The Awash Arba and Werer towns are located

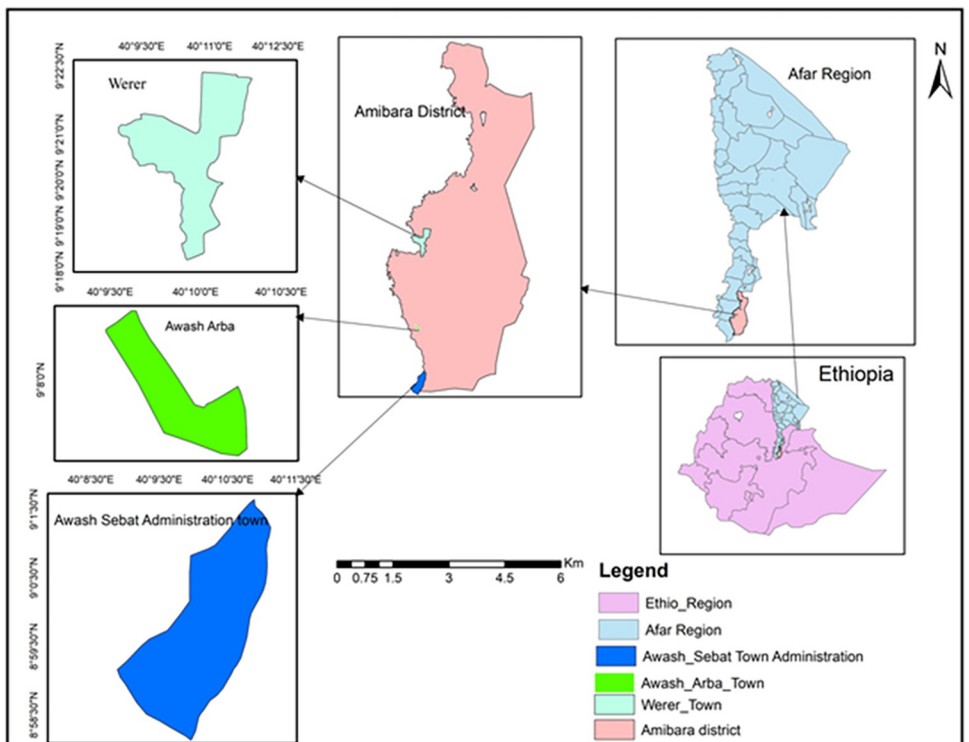

**Fig 1. Map of the study areas.** A map showing the locations of the three study areas: Awash Sebat, Awash Arba and Werer towns, in the Southern Afar Region, Ethiopia.

about 250 km to the north-east of Addis Ababa at 9.33453˚N latitude, 40.181385˚E longitude, and an altitude range of 720 to 1100 meters above sea level (masl) (Central Statistics Agency, 2019, unpublished). Awash Arba and Werer sites were selected from Amibara district, which is about 30 km apart (Fig 1). Agro-ecologically, these two towns are semi-arid, with temperatures ranging from 25˚C to 35˚C and an average annual rainfall of 530 mm. The communities are pastoralist and agro-pastoralist, in which they mainly obtain income from livestock rearing and other agricultural crop production [28].

Awash Sebat town administration is located 214 km north-east of Addis Ababa at 8.98810˚N latitude and 40.163936˚E longitude (Central Statistics Agency, 2019, unpublished). Its altitude ranges from 820 masl to 1120 masl, and the temperatures vary from 22.6˚C to 30.6˚C. The town and its surroundings receive a mean annual rainfall of 606.6 mm.

The study areas are found in the middle of the Awash Valley, where infectious diseases such as malaria, chikungunya, and dengue fever are common [29]. Considerable attention has been given by the Regional government to achieve better health in the region, including the three towns [30]. The Region has hospitals, health centers, clinics, and health posts in different zones and districts. There is one hospital, one clinic, five health centers, and a few private clinics. Insecticides such as deltamethrin, permethrin, alpha-cypermethrin, pirimiphos-methyl bendiocarb, and propoxur have been used for indoor residual spraying or pyrethroids in long-lasting insecticide nets for the control of malaria and arboviral diseases in these towns [26, 31]. Similarly, since maize, sorghum, and cotton productions are common in the area, agricultural chemicals such as soil fertilizers and agricultural pesticides are being used to increase crop production [32].

### Larval collection and rearing

A cross-sectional study was designed to collect larvae and pupae of *Aedes* species from July-October, 2022 in the Awash Sebat, Awash Arba, and Werer towns. The study towns were selected purposively in collaboration with the local district health professionals and on the basis of the recent outbreak reports of dengue fever and chikungunya virus (Awash Sebat town and Amibara district health bureaus, 2022, unpublished). The towns were surveyed once per month. Larvae and pupae were collected from different positive habitats, including tyres, water storage drums, discarded plastic and metallic containers, using standard dipper (350 ml), ladles, pipets, and nets. The larvae and pupae collected from each habitat were transferred into labelled plastic jars, transported to the insectary of the Aklilu Lemma Institute of Pathobiology, Addis Ababa University, Ethiopia, and reared to adults under standard laboratory conditions. Tetra cichlid floating fish food (®/TM/©2019 Germany) was provided as food to the larvae. The emergence of larvae into pupae was checked each morning, and pupae were collected in glass flasks and transferred into (30 cm x 30 cm x 30 cm) netted cages. Emerged adult *Ae. aegypti* were kept at 27 ± 2˚C temperature, 75 ± 10% relative humidity, and provided access to feed on a 10% sucrose solution until the bioassay tests. Emergent adults were identified by species using morphological taxonomic keys for *Aedes* species [33].

### Phenotypic susceptibility status of *Aedes aegypti*

Prior to insecticide susceptibility tests, sugar-fed females *Ae. aegypti* were starved for 2 hrs. Non-blood-fed, 3-5-old females *Ae. aegypti* mosquitoes were tested for their susceptibility to insecticides in tube test following the World Health Organization's insecticide testing procedures [20, 34]. They were exposed to pyrethroids (0.05% deltamethrin, 0.05% alpha-cypermethrin, and 0.75% permethrin), carbamates (0.1% bendiocarb and 0.1% propoxur), and organophosphate (0.25% pirimiphos-methyl) [34]. The insecticides were selected for the study because they are among the insecticide classes which have been used for the control of public health important mosquitoes in Ethiopia and the study areas. Similar concentrations of insecticides were used to test the susceptibility of *Ae. aegypti* in Tanzania [35]. However, we did not follow the World Health Organization's recommended discriminating concentrations for *Aedes* species due to the unavailability of insecticide-impregnated papers at the time of the experiment. This practical obstacle required a strategy that looked into the alternative doses of the insecticides. In each test, 20 females of *Ae. aegypti* were exposed to each of the five treatment replicates and two control test tubes, making a total of 140 mosquitoes in a single test. The exposed mosquitoes were observed for 60 minutes at an interval of 10 minutes to observe their knockdown. After an hour, the exposed females were transferred into holding tubes and kept for 24 hrs at 27.0 ± 2.0˚C temperature, 75 ± 10% relative humidity, and were offered cotton wool pads soaked in a 10% sugar solution. The mortality in mosquitoes was recorded 24 hrs after 60 minutes of exposure. *Ae. aegypti* susceptibility test was repeated for alpha-cypermethrin in Awash Arba town since *Ae. aegypti* populations appeared to be possibly phenotypic resistance to this insecticide. However, in Werer town, the susceptibility status of *Ae. aegypti* to bendiocarb was not conducted due to insufficient mosquito samples.

### Ethical issues

We obtained ethical approval from the Institutional Review Board (IRB) of the Aklilu Lemma Institute of Pathobiology, Addis Ababa University, Ethiopia (June 13, 2022; ALIPB IRB/80/2022). Written permission letters were obtained from the district health bureaus and health centers in the towns. The households were informed about the objective of the study, and their agreements were sought prior to the larvae and pupae collection.

## Data analysis

Mortality was determined on the basis of the dead and alive females of *Ae. aegypti* after the 24 hours recovery period. Susceptible were those with a mortality in the range of 98–100%, possibly or suspected for resistance were with a mortality 90–97%, and resistant if mortality was <90% [20]. Probit analysis was used to determine the estimated duration at which 50% (KDT$_{50}$) and 95% (KDT$_{95}$) of the exposed *Ae. aegypti* mosquitoes were knocked-down. Overall adult mortalities and knockdown times were computed using SPSS Software version 20.

## Results

### Knockdown effect of insecticides against *Aedes aegypti*

The knockdown times of *Ae. aegypti* exposed to insecticides are given in Table 1. The time to knockdown 50% (KDT$_{50}$) of *Ae. aegypti* exposed to deltamethrin ranged from 12.15 to 17.06 minutes whereas KDT$_{95}$ of the insecticide ranged from 18.74 to 34.01 minutes. The KDT$_{95}$, (34.01 minutes) observed for *Ae. aegypti* collected from Awash Sebat was found to be longer than for other towns (Table 1). On the other hand, *Ae. aegypti* exposed to permethrin showed the lowest KDT$_{50}$ in Awash Arba town (KDT$_{50}$, 5.11). Overall, the KDT$_{50}$ and KDT$_{95}$ of *Ae. aegypti* varied among *Ae. aegypti* across the study localities.

### Susceptibility status of *Aedes aegypti* to insecticides

The percentage mortalities of *Ae. aegypyi* mosquito populations against insecticides are presented in Figs 2–4. Accordingly, *Ae. aegypti* mosquitoes showed phenotypic resistance to bendiocarb and propoxur insecticides, with mortality rates of 88% and 87% respectively in Awash Sebat town. There was 100% mortality of *Ae. aegypti* to deltamethrin, permethrin, alpha-cypermethrin, and pirimiphos-methyl in Awash Sebat town (Fig 2). All *Ae. aegypti* in the control group showed 0% mortality in all study sites.

*Ae. aegypti* populations from Awash Arba were also observed to be resistant to bendiocarb and propoxur but were possibly resistant to alpha-cypermethrin (95% mortality) (Fig 3). *Ae. aegypti* populations were fully susceptible to deltamethrin, permethrin, and pirimiphos-methyl insecticides in this town.

**Table 1. The times (in minute) to knockdown of 50% and 95% of *Ae. aegypti* exposed to commonly used pyrethroids at three localities during July-October 2022 in Southern Afar region Ethiopia.**

| Insecticides | Localities | KDT$_{50}$ [95% CI] | KDT$_{95}$ [95% CI] |
|---|---|---|---|
| Deltamethrin | Awash sebat | 17.06 [15.81–18.28] | 34.01 [31.02–38.12] |
| | Awash Arba | 12.15 [11.41–12.97] | 18.74 [16.98–21.56] |
| | Werer | 16.53 [15.37–17.67] | 30.92 [28.24–34.62] |
| Permethrin | Awash Sebat | 8.92 [7.34–10.26] | 24.21 [21.23–28.93] |
| | Awash Arba | 5.11 [2.91–6.94] | 19.23 [16.23–24.09] |
| | Werer | 15.12 [12.45–17.66] | 29.89 [24.64–41.53] |
| Alpha-cypermethrin | Awash Sebat | 14.35 [13.35–15.35] | 24.93 [22.77–27.97] |
| | Awash Arba | 13.47 [12.64–14.36] | 20.78 [19.04–23.32] |
| | Werer | 16.71 [13.85–19.46] | 32.90 [27.26–44.98] |

CI, Confidence interval; KDT$_{50}$, time taken for 50% of the tested *Ae. aegypti* to be knocked-down; KDT$_{95}$ time taken for 95% of the tested *Ae. aegypti* to be knocked-down

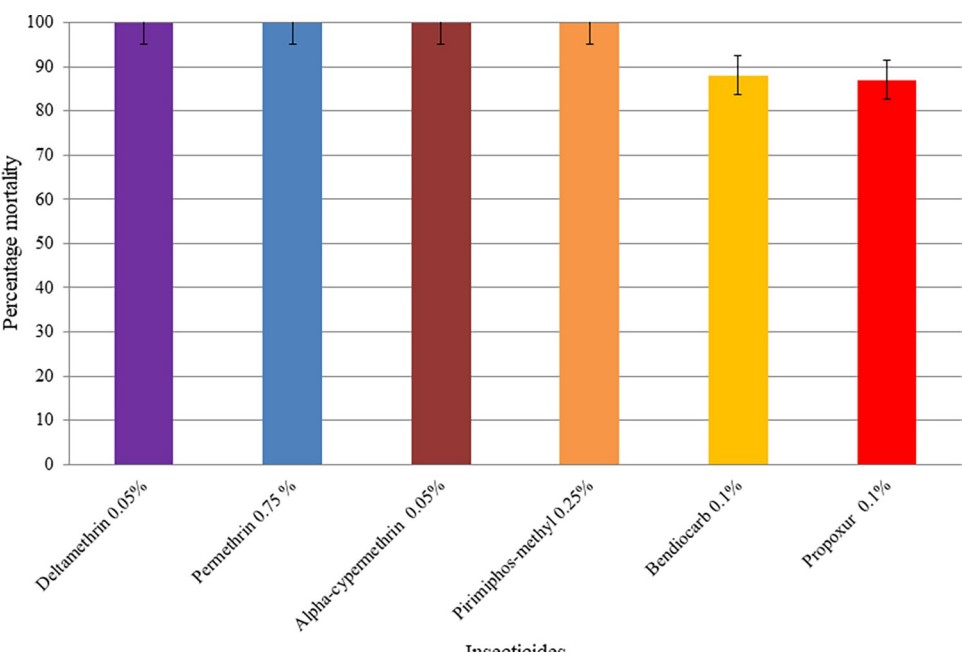

**Fig 2. Insecticide susceptibility status of *Ae. aegypti* populations from Awash Sebat town, Southern Afar region, Ethiopia, July-October 2022.** Error bars represent 95% confidence intervals.

Mortality in *Ae. aegypti* from Werer town was 87% with propoxur and 99% with alpha-cypermethrin (Fig 4). However, *Ae. aegypti* from this site was found to be fully susceptible to deltamethrin, permethrin, and pirimiphos-methyl.

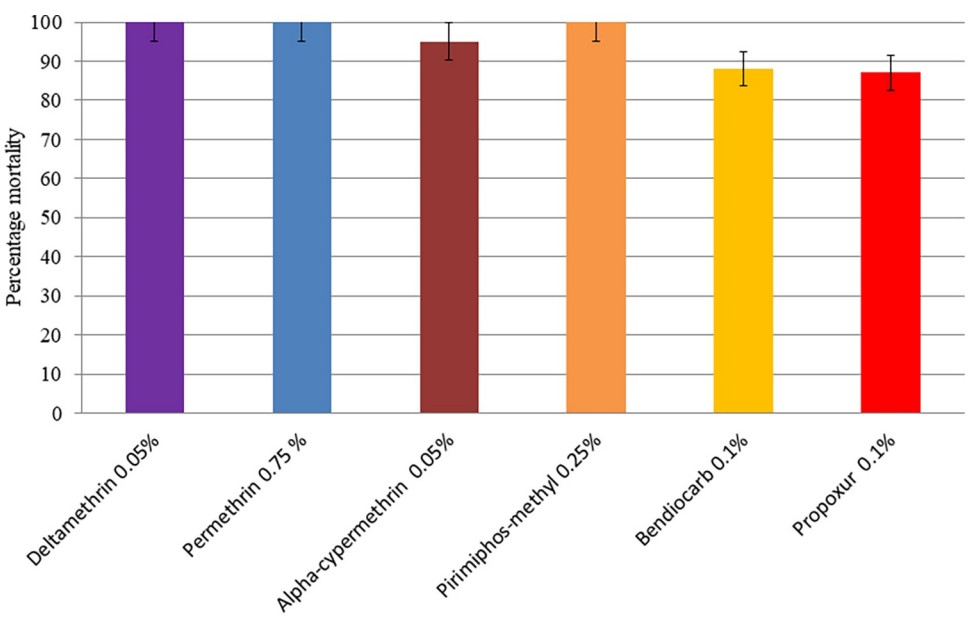

**Fig 3. Insecticide susceptibility status of *Ae. aegypti* populations from Awash Arba town, Southern Afar region, Ethiopia, July-October 2022.** Error bars represent 95% confidence intervals.

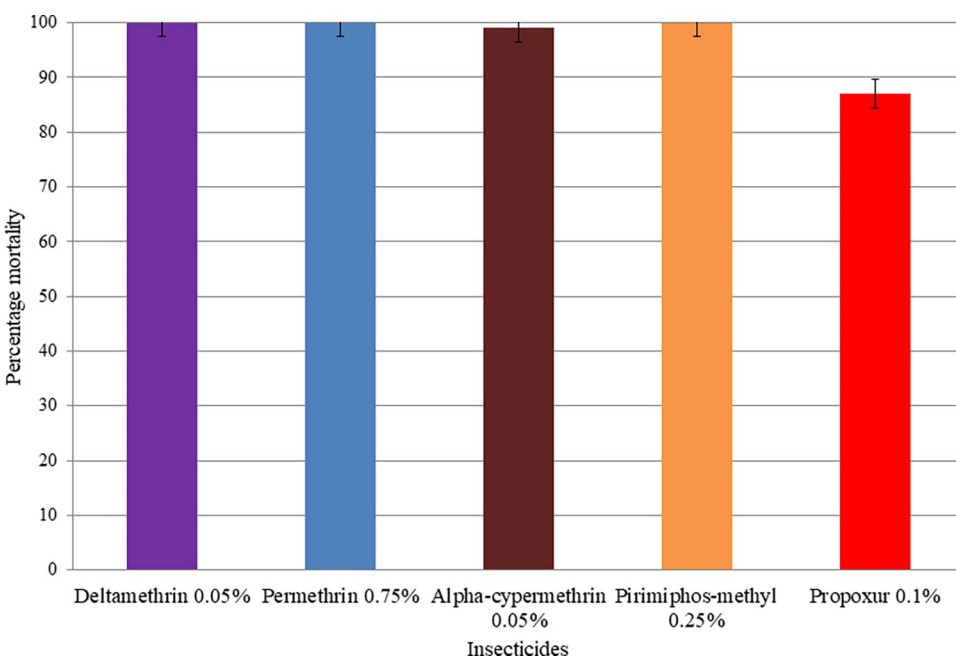

**Fig 4. Insecticide susceptibility status of *Ae. aegypti* populations in Werer town, Southern Afar region, Ethiopia, July-October 2022.** Error bars represent 95% confidence intervals.

## Discussion

Recent outbreaks of dengue fever and chikungunya virus fever in Eastern and the North-eastern parts of Ethiopia especially in Dire Dawa city administration, Somali Regional State, Amibara and Gewane districts of Afar Regional State require the control of the *Aedes* species that transmit the viruses. Insecticide-based mosquito control is underway in Ethiopia in general and in the Afar Regional State in particular [7, 36]. In so doing, there has been continuous evaluation of the efficacy of the insecticides against malaria-transmitting *Anopheles* species, but there is a scarcity of research findings conducted to control *Aedes* species either chemically, biologically, or by other means of vector control strategies in the country [37]. Thus, the control of the viral disease transmitting *Ae. aegypti* vector is poorly investigated and needs detailed research. As far as our knowledge is concerned, this study was the first in its kind to report the susceptibility status of *Ae. aegypti* to public health insecticides in the Southern Afar Regional State, and there are limited data to compare our findings to others in the country.

*Aedes aegypti* is considered by some authors to be highly exophilic [38], exophagic, and a daytime biter [39]. It usually feeds on human and other vertebrate hosts, as described in previous study [39]. However, it may tend to be endophilic and endophagic, in which case it rests and feeds indoors [40]. This makes control of *Ae. aegypti* and the *Aedes*-borne diseases it transmits more difficult [41]. As a result, different intervention tools have been adopted to control *Aedes* mosquitoes including indoor residual insecticide spraying and long-lasting insecticide treated nets in addition to managing their breeding habitats [42].

In the present study, *Ae. aegypti* showed higher $KDT_{50}$ and $KDT_{95}$ as compared to the study conducted in Sudan on deltamethrin insecticide. For instance, the $KDT_{50}$ and $KDT_{95}$ observed for deltamethrin in Sudan were 13 and 19 minutes, respectively which were lower than the current study in Awash Sebat and Werer towns [43]. However, the study conducted in Nigeria revealed that the $KDT_{50}$ of deltamethrin was almost twice the $KDT_{50}$ observed in Awash Sebat [44]. A relatively similar permethrin $KDT_{50}$ was observed in Awash Sebat town

and Mlabani, Tanzania. However, higher $KDT_{95}$ with deltamethrin and permethrin was found for Mlabani than all the present study towns [35]. Overall, the tube test revealed that *Ae. aegypti* was susceptible to deltamethrin in Sudan and Tanzania but possibly resistant in Nigeria.

*Aedes aegypti* was found to show varying susceptibility to different insecticides. *Ae. aegypti* populations were observed to be phenotypically resistant to bendiocarb with < 90% mortality. *Ae. aegypti* resistance to bendiocarb was also reported in the previous study in Tanzania during the rainy season [35]. However, this finding was contrary to the findings observed from Sudan, Nigeria, and Cambodia [43–45]. The World Health Organization has recommended in 2022 that the insecticide discriminating dose of bendiocarb for *Ae. aegypti* be changed from its initial 0.1% to 0.2% [34]. Thus, future efficacy studies of bendiocarb should target the 0.2% concentration [34].

In the present study, *Ae. aegypti* was found to be resistant to propoxur 0.1% concentration in all the study towns. Similar finding was reported in Thailand, where the efficacy of *Ae. aegypti* mortality to propoxur was 83%, which indicated the phenotypic resistance of *Ae. aegypti* to propoxur in this area [46].

The possible contributing factors to the emergence of phenotypic resistance of *Ae. aegypti* in the present study areas could be frequent uses of insecticide treated bed nets and indoor residual insecticide spraying to control mosquito-borne diseases as previously indicated [47]. On the other hand, Ethiopia has had a long history of utilizing pesticides to increase agricultural production and improve human health since 1964 [48, 49]. Unfortunately, however, agricultural pesticides induce insecticides resistance selection pressure in different ways. For instance, pesticides used for agriculture pest control can select vector control insecticides because of a similar mode of action, and bio-active ingredients might drive a section pressure for chemical based insecticide control tools [27]. Similarly, agro-pesticides used in agriculture may increase insecticide tolerance as the pesticides are washed into mosquito breeding habitats, which confer resistance mechanisms at the adult stage [50]. Thus, the emergence of insecticide resistance in bendiocarb and propoxur insecticides in the present study could potentially arise from the frequent exposure of the mosquito to carbamate insecticides in agriculture used as agricultural pesticides or herbicides. In addition, *Ae. aegpti* resistance to propoxur and bendiocarb may occur as a result of mutations in the insecticide target site enzyme Acetyl-cholinesterase (AChE) as observed in Cuba [51].

Pyerethroids such as deltamethrin and alpha-cypermethrin are commonly used to control malaria and other mosquito-borne diseases in Ethiopia. In the present study, the pyrethroid alpha-cypermethrin showed less than 97% of the mortality of adult *Ae. aegypti* populations in the Awash Arba town, which revealed the emergence of possibly resistance in *Ae. aegypti* in this area. This might be due to the frequent use of pyrethroids including alpha-cypermethrin in the insecticide-treated nets to control malaria and agricultural pesticides in this area as discussed above. This may lead to failures in vector control intervention programs in the study area and elsewhere in the country if resistance management is not undertaken. Similar findings were reported in Ghana [23].

Completely/fully susceptible *Ae. aegypti* populations were observed to the organophosphate, i.e., pirimiphos-methyl in all the study areas. Likewise, the study conducted in Cambodia [45] reported similar findings in that *Ae. aegypti* populations were completely susceptible to pirimiphos-methyl even in its lower concentration than the present study. However, this finding was contrary to the finding reported in Nigeria [44]. Similarly, *Ae. aegypti* was also fully susceptible to deltamethrin and permethrin in the study areas. These findings were contrary to the findings reported in Malaysia [52], where resistance phenotypes were observed in deltamethrin and permethrin. However, similar findings were reported in Tanzania [35] and

in Peru [53], where deltamethrin showed greater than 99% of mortality in adult *Ae. aegypti*. Thus, these findings could serve as a baseline for the Ethiopian Ministry of Health and the Afar Region Health Bureau to implement chemical interventions, particularly the pyrethroids deltamethrin, permethrin, and organophosphate pirimiphos-methyl, either in the form of aerial spray or insecticidal surface treatment, to reduce the density of *Ae. aegypti* and prevent the burden of arboviral diseases during disease outbreaks in different parts of the country.

## Conclusion

The present study showed that *Ae. aegypti* populations were resistant to bendiocarb and propoxur, possibly resistant to alpha-cypermethrin, and fully susceptible to deltamethrin, permethrin, and pirimiphos-methyl insecticides. Thus, based on the findings, regular monitoring of insecticide resistance should be carried out in the study areas and elsewhere in Ethiopia, including molecular aspects, to mark and evaluate resistance patterns. Moreover, proper use of existing insecticides is mandatory to prevent *Ae. aegypti* resistance to insecticides in the study areas. In addition, the organophosphate pirimiphos-methyl can be used in addition to deltamethrin and permethrin in the control of adult *Ae. aegypti* mosquitoes in Ethiopia.

## Supporting information

**S1 Table. Observed knockdown times (in minutes) and number of knocked-down female *Ae. aegypti* exposed to insecticides during the 60 minutes in Awash Sebat, Awash Arba and Werer towns of Southern Afar, July-October 2022.**
(DOCX)

**S2 Table. Mortality of *Ae. aegypti* 24 hrs, after 1 hr exposed to insecticides Awash Sebat, Awash Arba and Werer towns, Southern Afar Regional State, Ethiopia, July-October, 2022.**
(DOCX)

**S3 Table. Representative GPS points of the Awash Sebat, Awash Arba and Werer towns of Southern Afar Regional State, Ethiopia recorded during *Ae. aegypti* larvae and pupae collection.**
(DOCX)

## Acknowledgments

The authors would like to acknowledge the Aklilu Lemma Institute of Pathobiology, Addis Ababa University, Ethiopia, for providing us with field and laboratory materials and insecticide impregnated papers. We would also like to thank the vector biology and control unit staffs, Mr. Wosen Sisay, for his technical assistance in the insectary. Finally, we appreciated all the communities and data collectors from Awash Sebat, Awash Arba, and Werer towns, for their collaboration during the study period.

## Author Contributions

**Conceptualization:** Mohammed Seid, Abebe Animut.

**Data curation:** Mohammed Seid.

**Formal analysis:** Mohammed Seid.

**Funding acquisition:** Mohammed Seid, Abebe Animut.

**Investigation:** Mohammed Seid, Esayas Aklilu, Abebe Animut.

**Methodology:** Mohammed Seid, Esayas Aklilu, Abebe Animut.

**Supervision:** Mohammed Seid, Esayas Aklilu, Abebe Animut.

**Validation:** Mohammed Seid, Esayas Aklilu, Abebe Animut.

**Visualization:** Mohammed Seid, Esayas Aklilu, Abebe Animut.

**Writing – original draft:** Mohammed Seid.

**Writing – review & editing:** Mohammed Seid, Esayas Aklilu, Abebe Animut.

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
