## [Decision Letter · Decision Letter 0]

29 Feb 2024

PONE-D-24-03915Susceptibility status of Aedes aegypti (Diptera: Culicidae) to public health insecticides in Southern Afar region, EthiopiaPLOS ONE

Dear Dr. Seid,

Thank you for submitting your manuscript to PLOS ONE. After careful consideration, we feel that it has merit but does not fully meet PLOS ONE’s publication criteria as it currently stands. Therefore, we invite you to submit a revised version of the manuscript that addresses the points raised during the review process. Please submit your revised manuscript by Apr 14 2024 11:59PM. If you will need more time than this to complete your revisions, please reply to this message or contact the journal office at plosone@plos.org. Please include the following items when submitting your revised manuscript:A rebuttal letter that responds to each point raised by the academic editor and reviewer(s). You should upload this letter as a separate file labeled 'Response to Reviewers'.A marked-up copy of your manuscript that highlights changes made to the original version. You should upload this as a separate file labeled 'Revised Manuscript with Track Changes'.An unmarked version of your revised paper without tracked changes. You should upload this as a separate file labeled 'Manuscript'.

We look forward to receiving your revised manuscript.

Kind regards,

James Colborn

Academic Editor

PLOS ONE

Journal Requirements:

2. Please note that funding information should not appear in the Acknowledgments section or other areas of your manuscript. We will only publish funding information present in the Funding Statement section of the online submission form. Please remove any funding-related text from the manuscript. 

3. In this instance it seems there may be acceptable restrictions in place that prevent the public sharing of your minimal data. However, in line with our goal of ensuring long-term data availability to all interested researchers, PLOS’ Data Policy states that authors cannot be the sole named individuals responsible for ensuring data access (http://journals.plos.org/plosone/s/data-availability#loc-acceptable-data-sharing-methods).

4. Please ensure that you refer to Figure 1 in your text as, if accepted, production will need this reference to link the reader to the figure.

5. We note that Figure 1 in your submission contain map images which may be copyrighted. All PLOS content is published under the Creative Commons Attribution License (CC BY 4.0), which means that the manuscript, images, and Supporting Information files will be freely available online, and any third party is permitted to access, download, copy, distribute, and use these materials in any way, even commercially, with proper attribution. For these reasons, we cannot publish previously copyrighted maps or satellite images created using proprietary data, such as Google software (Google Maps, Street View, and Earth). For more information, see our copyright guidelines: http://journals.plos.org/plosone/s/licenses-and-copyright.

(1) You may seek permission from the original copyright holder of Figure 1 to publish the content specifically under the CC BY 4.0 license.  

**Additional Editor Comments:**

All reviewers in felt that the manuscript lacked sufficient detail to fully evaluate. In particular they requested more detail on the methodology, including the sampling methods and statistical analyses performed. They have also listed additional specific changes that should be addressed prior to resubmission.

Reviewers' comments:

Reviewer's Responses to Questions

**Comments to the Author**

1. Is the manuscript technically sound, and do the data support the conclusions?

Reviewer #1: Yes

Reviewer #2: Yes

Reviewer #3: No

2. Has the statistical analysis been performed appropriately and rigorously? 

Reviewer #1: No

Reviewer #2: Yes

Reviewer #3: No

3. Have the authors made all data underlying the findings in their manuscript fully available?

Reviewer #1: Yes

Reviewer #2: No

Reviewer #3: No

4. Is the manuscript presented in an intelligible fashion and written in standard English?

Reviewer #1: Yes

Reviewer #2: No

Reviewer #3: Yes

5. Review Comments to the Author

Reviewer #1: The authors conducted a study to evaluate the susceptibility status of Aedes aegypti (Diptera: Culicidae) to public health insecticides in the Southern Afar region, Ethiopia. The findings revealed that populations of Ae. aegypti in the study area showed phenotypic resistance to carbamates such as propoxur and bendiocarb. However, they were susceptible to certain pyrethroids, including permethrin, alphacypermethrin, and deltamethrin, as well as the organophosphate insecticide primiphos-methyl. The study provides compelling evidence of an emergence of insecticide resistance within Aedes aegypti populations in the country. While the study provides valuable insights into the emergence of insecticide resistance within Aedes aegypti populations, the writing of the paper requires revision to enhance its overall quality.

Major comments

1. What criteria underlie the selection of insecticides in this study, specifically the inclusion of three pyrethroids and two carbamates? Additionally, are these insecticides currently in use for public health purposes?

2. According to the WHO guideline cited in the methodology section, the recommended discriminating doses for susceptibility tests of Aedes mosquitoes using pyrethroids are 0.25% Permethrin, 0.03% alpha-cypermethrin, and 0.03% deltamethrin. However, it appears that different discriminating doses of pyrethroids were used for the study. Why it is different?

3. Why only phenotypic resistance studied? Was there a specific reason for not investigating the resistance mechanisms at play in the mosquito populations?

4. How did you identified the Aedes mosquitoes used for the susceptibility tests were all Ae. Aegypti?

5. How and why was the association between pyrethroid resistance in the study area and agricultural practices established?

Minor comments

1. Are Ae aegypti and Ae bromeliae the only two species these the only two Aedes species existing in the country? For example Ae. vittaus and Ae. africanus also exist in the country.

2. Why the Knockdown effect of permethrin and alpha-cypermethrin not elaborated like that of deltamethrin?

3. The vertical axis maximum value should not exceed 100 (Fig 2). This works for all graphs

4. The susceptibility test of alphacypermethri in Awash Arba town repeated or not?

5. Why is integrated vector control strategies recommended without supporting data from the study findings?

6. How are pyrethroids currently utilized in the control of vectors?

Suggestions

Suggestions and comments are provided in the main document

Reviewer #2: In recent years arboviral diseases have emerged as a major public health problem in Africa, calling attention for implementing effective disease control approaches, especially the insecticide-based interventions. Effective chemical interventions require an up-to-date knowledge of insecticide resistance in vector species, therefore, this study is very relevant for Ethiopia.

I suggest improvement in the manuscript and offer the following comments and suggestions:

Overall: correct numerous typos and improve English language; ensure the references cited in the text match with those in the References list.

Abstract:

- Background: In the last sentence of the “Background”, mention names of the insecticides instead of stating “public health insecticides”.

- Methodology: Modify the sentence “Mortality was recorded after 24 hours” to “The mortality in mosquitoes was recorded 24 hours after 60 minutes of exposure”. Delete the abbreviation “KDT” for knockdown rate since later you have used KDT for the knockdown time.

- Results: In this sentence KDT95 implies Knockdown time, so modify the sentence to “Aedes aegypti from Awash Arba showed high knock down time (KDT95 26.5 minutes) to deltamethrin ….”.

- Results: Rename “alphacypermethrin” to “alpha-cypermethrin”, “delthamethrin” to “deltamethrin” and “pirimiphos methyl” to “pirimiphos-methyl” throughout the manuscript (in text, figures, and tables).

- Conclusion: delete “control” at the end of the second sentence. Mention the insecticide discriminating concentration for each insecticide impregnated paper used. Also state that further studies are required to confirm resistance to alpha-cypermethrin.

Introduction

- Correct typos: change “Ae. albopictus[11, 12].” to “Ae. albopictus [11, 12].”, “Ae .aegypti” to “Ae. aegypti”, “Fenitrothion” to “fenitrothion”.

Material and Methods

Study sites:

- Correct many typos (e.g. “820masl” to “820 masl” and so on.

- Exactly what insecticide spraying is practiced in the study areas for the control of malaria and arboviral diseases to give an indication of insecticidal interventions used.

- Figure 1 is not mentioned in the text under the study sites.

Insecticide susceptibility:

- Reference # 16 attributed to the WHO procedure is incorrect. Please recheck correctness of all references used in the text.

- Correct numerous typos such as “delta-methrin” to “deltamethrin”.

- A correct number of mosquitoes (100) were exposed in the treatments but not in the control (40 instead of 50). This is a slight deviation from the WHO tube test procedure.

- “fed on 10% sugar solution” is an incorrect statement; instead, state “were offered cotton wool pads soaked in 10% sugar solution”.

Data analysis

- Correct this statement “suspected for resistance were with a mortality 90-98%” to “suspected of resistance were with a mortality of 90–97%”.

- Reference # 34 is incorrect; the correct reference should be: WHO (2022). Manual for monitoring insecticide resistance in mosquito vectors and selecting appropriate interventions. https://iris.who.int/handle/10665/356964.

- Correct the abbreviations “50% (KD50) and 95% (KD95)” to “50% (KDT50) and 95% (KDT95).

Results

- In Table 1 header, change “KDT50±SE” to “KDT50 ± SE” and “KDT95±SE” to “KDT95 ± SE”

- Correct numerous typos in the text and figures.

Discussion

- Is there any publication supporting the following statement “Ae. aegypti is generally considered as exophilic, exophagic …....”.

- While KDT95 for deltamethrin in one town was higher, the tube test does confirm full susceptibility of Ae. aegypti.

- Some of the WHO insecticide discriminating concentrations for Aedes aegypti adults have changed in 2022 following the publication of the above said WHO manual based on a WHO multi-centre study in which the authors’ institution was a collaborator, as follows:

o deltamethrin 0.05% to 0.03%

o permethrin 0.75% to 0.40%

o bendiocarb 0.1% to 0.2%

o pirimiphos-methyl 0.25% to 60 mg/m2

o alpha-cypermethrin 0.05% : no change

o propoxur 0.1% : no change

Since Ae. aegypti was found resistant to bendiocarb 0.1% concentration while its concentration has been increased to 0.2%, discuss that further tests should be conducted with 0.2% impregnated papers to confirm resistance.

References

- I did not check correctness of each reference cited, but I noticed at least two incorrect references, therefore authors should recheck all references cited.

Reviewer #3: The authors have a good reason for conducting this research, the manuscript is well organized and written clearly enough to be accessible to non-specialists. However, lacks a detailed description of the method used to collect larvae and pupae. It is understood that they collected in three places in the South of the Afar region of Ethiopia, where vector-transmitted infections occur, and that they did so from July to October 2022. However, they do not specify how many times each of the locations were visited during that 3 to 4 months for the collections to obtain a variety of samples, nor how many breeding sites they found from where they collected the specimens. The latter is important because it is not known if the offspring they collected come from the same progeny or from few of them, hence the importance of collecting in different breeding sites and at different collection times in the same place. The material collected on the same day at various breeding sites of a locality must be combined so that there is mixing of the genes and thus the results of resistance to insecticides may correspond to several parental mosquitoes originated from the collected site. It is important to describe the number of samples collected on the dates in each of the locations. If possible, describe the number of larvae and pupae collected or an approximate number.

By other hand, it is not specified how many experiments were carried out for the tests with each insecticide. It is understood that only 100 mosquitoes were used and 40 for two control tubes, so that makes only a test. However, at least two other replicates of collections separated in time must be carried out for the same collection location. In this way you can work with an average mortality for more precise data on the locality that is being collected, and if diferences in mortalities are found in the three collections could be a good motive for discussion in this manuscript to enrich the discussion.

Discussion, 3rd paragraph, at the end: reffering to the phrase "the findings observed in the present study were lower than results reported in Sudan, Tanzania and Nigeria" it would be good to add if those from Sudan, Tanzania and Nigeria were reported as resistant or only tolerants to deltamethrin, and maybe is good to mention their KDT50 and 95 values, so we can have an idea for comparison.

A sentence could be added in this section about the resistance mechanisms that could be present in the mosquitoes detected resistant to propoxur and bendiocarb, but not to organophosphate. It is known that the two groups of insecticides share the same site of action, and that if resistance to both groups has been diagnosed, such a situation could explain a resistance based on a point mutation in the site of action, however this is not the case.

Minor corrections:

Introduction, 3rd paragraph. "Fenitrothion" cjhange to "fenitrothion"

Material and Methods: 1st paragraph. "Were" change to "Werer"

Insecticide susceptibility paragraph: "delta-methrin" change to "deltamethrin" and "bendio carb" change to bendiocarb"

Results, 1st paragraph, Table 1. "Ae. Aegypti" change to "Ae. aegypti"

Discussion, 4th paragraph. "propxur" change to "propoxur"

Conclusion. "perimethrin" change to "permethrin"

Check the double spaces between words throughout the writing and correct them to one space.

6. PLOS authors have the option to publish the peer review history of their article (what does this mean?). If published, this will include your full peer review and any attached files.

Reviewer #1: **Yes: **Eba A.Simma

Reviewer #2: **Yes: **Dr Rajpal Singh Yadav

Reviewer #3: No

---

## [Author Response · Author response to Decision Letter 0]

29 Apr 2024

Comments on journal requirements

1. Follow the PLOS ONE Journal submission guidelines.

Response: We have followed PLOS ONE Journal submission guidelines throughout the revised manuscript.

2. Funding information: 

Response: we have provided the funding information. The statement “the study was financially supported by Addis Ababa University the Vice President for Research and Technology Transfer Office through its thematic research grant” has been included. Unfortunately, we did not include the funding statement in the submission system before. Thus, we kindly request you to incorporate the funding statement on the online submission system.

3. Non-author contact information: 

Response: To ensure guarantee for the long-term stability and availability of the data, we provided the data used as supporting information file 1, supporting information file 2 and supporting information file 3. Supporting information file 1 shows measurement used to estimate duration in minutes in which KDT50 and KDT95 Aedes aegypti knocked-down and the 95% confidence interval were analyzed (Table 1 in the manuscript). Supporting information file 2 presents mortality values of Ae. aegypti used to build the percentage mortality graphs (graphs 2-4 in the manuscript). Supporting information file 3 reveals representative GPS points of the study sites recorded during Ae. aegypti larvae and pupae collection. We mapped the study areas using Ethio-region GIS shape files to show the location of the study sites. We did not use map image for analysis in the result section of the manuscript. 

4. Figure 1 copy-right: 

Response: Authors have mapped the study area (named as Figure 1 in the manuscript) using arc GIS software version 10.8 from GIS Shape files of Ethiopia freely and no individual or organization involved. Thus, there is no copy-right issue for this figure. 

Additional Editor’s comments

All reviewers in felt that the manuscript lacked sufficient detail to evaluate. In particular they requested more detail on the methodology including the sampling methods and statistical analyses performed. They have also listed additional specific changes that should be addressed prior to resubmission. 

Response: We have provided details on each section as per the comments given by the reviewers. We have made slight modification where necessary in the revised manuscript. 

Reviewer #1 

Major comments

1. What criteria underlie the selection of insecticides in this study, specifically the inclusion of three pyrethroids and two carbamates? Additionally, are these insecticides currently in use for public health purposes?

Response: Viral diseases outbreaks are continuously reported in Ethiopia including the present study areas. Insecticide based viral diseases transmitting mosquito control is a quick responses during outbreaks. Thus, “the insecticides were selected for the study because they have been used for the control of mosquitoes of public health importance in Ethiopia and the study area”. “The insecticides are also among the insecticide classes used in the country”. The statements have been included in the methods (phenotypic susceptibility of Aedes aegypti) sub-section of revised manuscript. Yes, most are being used depending on the local transmission situation. 

2. According to the WHO guideline cited in the methodology section, the recommended discriminating doses for susceptibility tests of Aedes mosquitoes using pyrethroids are 0.25% Permethrin, 0.03% alpha-cypermethrin, and 0.03% deltamethrin. However, it appears that different discriminating doses of pyrethroids were used for the study. Why it is different? Response: Thank you for the important question. The insecticides are used for public health in the study areas in particular and in Ethiopia in general with the same concentration of the present study tests. Thus, we used concentrations of the insecticides, WHO criteria, used to evaluate insecticide susceptibility status of Anopheles arabiensis. Furthermore, studies from Tanzania and Burkina Faso were used similar concentration of insecticides to test Aedes aegypti susceptibility to insecticides (Kahamba et al., 2020; Namountougou et al., 2020). 

3. Why only phenotypic resistance studied? Was there a specific reason for not investigating the resistance mechanisms at play in the mosquito populations? 

Responses: We share the comment as a limitation. We were not able to detect resistant genes to bendiocarb and propoxur due to lack of the reagents. We recommend further molecular investigations regarding the resistant genes. 

4. How did you identified the Aedes mosquitoes used for the susceptibility tests were all Ae. Aegypti? 

Responses: Experienced personnel were carried out the collection and insecticide susceptibility tests. Selected numbers of the mosquitoes were identified before the test as Ae. aegypti. All the mosquitoes were checked for being Ae. aegypti at the end of each experiment. All the mosquitoes were identified as Ae. aegypti. 

5. How and why was the association between pyrethroid resistance in the study area and agricultural practices established?

Responses: Pyrethroids are mainly used in the control of mosquitoes that transmit viral diseases and malaria. They are also used in the control of agricultural pests (Shafer et al., 2008; Chrustek et al., 2018). It is known that insecticides used in agricultural practices induce public health insects to develop resistance against public health insecticides. Because some insecticides used in agricultural insects share a similar mechanism of action and bio-active ingredients with public health insecticides. In the study areas, farmers used the pyrethroids to control pests of crops. As a result, uses of the insecticides in agriculture might potentially contribute to resistant mosquitoes. 

Miner comments

1. Are Ae. aegypti and Ae. bromeliae the only two species these the only two Aedes species existing in the country? For example Ae. vittaus and Ae. africanus also exist in the country. 

Responses: Thank you for your relevant comment. The statement “In Ethiopia, Ae. aegypti, Ae. bromeliae, Ae. vittatus, Ae. hirsutus, Ae. simpsoni complex and Ae. africanus are implicated as the potential vectors of viral diseases” has been included in the revised manuscript as suggested. 

2. Why the Knockdown effect of permethrin and alpha-cypermethrin not elaborated like that of deltamethrin? 

Responses: The statement “On the other, Ae. aegypti exposed to permethrin showed the lowest KDT50 in Awash Arba”, has been included in the revised manuscript as commented.

3. The vertical axis maximum value should not exceed 100 (Fig 2). This works for all graphs. 

Responses: We have corrected the vertical axis of all the graphs to 100 in the revised manuscript as suggested.

4. The susceptibility test of alpha-cypermethrin in Awash Arba town repeated or not? 

 Response: Yes, the test was repeated. The statement “Ae. aegypti susceptibility test was repeated for alpha-cypermethrin in Awash Arba town since Ae. aegypti populations were appeared to be suspected phenotypic resistance to this insecticide” has been included in the revised manuscript.

5. Why is integrated vector control strategies recommended without supporting data from the study findings? 

Responses: Thank you for the relevant question. We modified the conclusion based on the findings of the study in the revised manuscript. 

6. How are pyrethroids currently utilized in the control of vectors?

Response: Pyrethroids are utilized currently to control mosquito vectors in the form of long lasting insecticides treated nets or in the insecticide residual spray. Moreover, recently the mosquito vector emerge resistance to pyrethroids. Thus, potential new approaches existing pyrethroids are emerging such as pyrethroids plus PBO (pyrethroid synergist piperonyl butoxide) which are used to break the pyrethroid resistance by blocking metabolic resistance. The PBO are not acting as insecticides but they facilitate the action of the pyrethroids.

Suggestions:

Response: All the suggestions given have been included in the revised manuscript. 

Reviewer #2 

I suggest improvement in the manuscript and offer the following comments and suggestions:

Overall: correct numerous typos and improve English language; ensure the references cited in the text match with those in the References list.

Response: The grammar and typos have been checked and corrected in the revised the manuscript. Similarly, all references cited in the text have been checked their correctness in the reference list. 

Abstract:

- Background: In the last sentence of the “Background”, mention names of the insecticides instead of stating “public health insecticides”.

Response: “deltamethrin, permethrin, alpha-cypermethrin, pirimiphos-methyl, bendiocarb and propoxur” have been included in the revised manuscript in the last sentence of Abstract’s background.

 -Methodology: Modify the sentence “Mortality was recorded after 24 hours” to “The mortality in mosquitoes was recorded 24 hours after 60 minutes of exposure”. Delete the abbreviation “KDT” for knockdown rate since later you have used KDT for the knockdown time.

Response: “The abbreviation “KDT” has been deleted. The mortality in mosquitoes was recorded 24 hours after 60 minutes of exposure” has been added in revised manuscript as suggested.

 Results: In this sentence KDT95 implies Knockdown time, so modify the sentence to “Aedes aegypti from Awash Sebat showed high knock down time (KDT95 34.01 minutes) to deltamethrin ….”

Response: “Deltamethrin in Ae. aegypti showed high knockdown time (KDT95 34.01 minutes) in Awash Sebat as compared to other towns has been rephrased.

-Results: Rename “alphacypermethrin” to “alpha-cypermethrin”, “delthamethrin” to “deltamethrin” and “pirimiphos methyl” to “pirimiphos-methyl” throughout the manuscript (in text, figures, and tables).

Response: “alphacypermethrin” to “alpha cypermethrin”, “delthamethrin” to “deltamethrin”, “pirimiphosmethyl” to “pirimiphos-methyl” have been renamed in the entire revised manuscript sections as suggested.

- Conclusion: delete “control” at the end of the second sentence. Mention the insecticide discriminating concentration for each insecticide impregnated paper used. Also state that further studies are required to confirm resistance to alpha-cypermethrin.

Response: The word control has been deleted from the conclusion. In addition, the statement “However, further studies should be carried out to evaluate the susceptibility of Ae. aegypti to alpha-cypermethrin insecticide in the Awash Arba area”, has been included in the revised manuscript.

-Introduction

-Correct typos: change “Ae. albopictus[11, 12].” to “Ae. albopictus [11, 12].”, “Ae .aegypti” to “Ae. aegypti”, “Fenitrothion” to “fenitrothion”.

Response: “Ae. albopictus[11, 12],” has been corrected to “Ae. albopictus [11, 12].” Ae .aegypti” to “Ae. aegypti”, “Fenitrothion” to “fenitrothion”, in the revised the manuscript as suggested.

-Material and Methods

Study sites: Correct many typos (e.g. “820masl” to “820 masl” and so on.

Response: “820masl” to “820 masl”, “22.6oC to 30.6oC” to “22.6 oC to 30.6 oC” and “606.6mm” to “606.6 mm” have been corrected in the revised manuscript.

 Exactly what insecticide spraying is practiced in the study areas for the control of malaria and arboviral diseases to give an indication of insecticidal interventions used.

Response: The statement “Insecticide such as deltamethrin, permethrin, alpha-cypermethrin, pirimiphos-methyl bendiocarb and propoxur are used,” has been included in the revised version of the manuscript.

 Figure 1 is not mentioned in the text under the study sites.

Response: Fig 1, has been cited in the text in the revised manuscript.

Insecticide susceptibility:

- Reference # 16 attributed to the WHO procedure is incorrect. Please recheck correctness of all references used in the text.

Response: All the references have been checked and reference number 16 has been corrected by reference 20 in the revised manuscript.

- Correct numerous typos such as “delta-methrin” to “deltamethrin”.

Response: delta-methrin” has been corrected to “deltamethrin in the revised manuscript.

- A correct number of mosquitoes (100) were exposed in the treatments but not in the control (40 instead of 50). This is a slight deviation from the WHO tube test procedure.

Response: We followed the WHO 2016 insecticide susceptibility test and exposed the control mosquitoes following the procedures in that manual.

- “fed on 10% sugar solution” is an incorrect statement; instead, state “were offered cotton wool pads soaked in 10% sugar solution”.

Response: the statement “were offered cotton wool pads soaked in 10% sugar solution” has been corrected in the revised manuscript as suggested.

Data analysis

 Correct this statement “suspected for resistance were with a mortality 90-97%” to “suspected of resistance were with a mortality of 90–97%”.

Response: the statement “suspected of resistance were with a mortality of 90–97%” has been added in the revised manuscript.

 Reference # 34 is incorrect; the correct reference should be: WHO (2022). Manual for monitoring insecticide resistance in mosquito vectors and selecting appropriate interventions. https://iris.who.int/handle/10665/356964.

Response: Since we initially used WHO (2016). Monitoring and managing insecticide resistance in Aedes mosquito populations: interim guidance for entomologists. Thus reference # 34 has been corrected by reference 20. 

 -Correct the abbreviations “50% (KD50) and 95% (KD95)” to “50% (KDT50) and 95% (KDT95).

Response: The abbreviations “50% (KD50) and 95% (KD95)” have been corrected to “50% (KDT50) and 95% (KDT95) in the revised manuscript as suggested.

-Results

-In Table 1 header, change “KDT50±SE” to “KDT50 ± SE” and “KDT95±SE” to “KDT95 ± SE”

Response: We have used estimated “KDT50 “KDT95 with their 95% confidence interval in the revised manuscript.

- Correct numerous typos in the text and figures.

Response: we have edited typos in the text and corrected the vertical axis of the figures in the revised manuscript.

-Discussion:

 -Is there any publication supporting the following statement “Ae. aegypti is generally considered as exophilic, exophagic …....”.

Response: reference number 37 cited in this study support the high exophilic and exophagic-ness of Ae. aegypti.

- While KDT95 for deltamethrin in one town was higher, the tube test does confirm full susceptibility of Ae. aegypti.

Response: The KDT95 for deltamethrin in Awash Sebat higher as compared to other pyrethroids in the study. Thus, it did not necessarily confirm resistance or suspect resistance since the mortality of Ae. aegypti were determined after 24 hrs. 

- Some of the WHO insecticide discriminating concentrations for Aedes aegypti adults have changed in 2022 following the publication of the above said WHO manual based on a WHO multi-centre study in which the authors’ institution was a collaborator, as follows:

o deltamethrin 0.05% to 0.03%

o permethrin 0.75% to 0.40%

o bendiocarb 0.1% to 0.2%

o pirimiphos-methyl 0.25% to 60 mg/m2

o alpha-cypermethrin 0.05% : no change

o propoxur 0.1% : no change

-Since Ae. aegypti was found resistant to bendiocarb 0.1% concentration while its concentration has been increased to 0.2%, discuss that further tests should be conducted with 0.2% impregnated papers to confirm resistance.

Response: the statements “World health organization in 2022 has recommended that the insecticide discriminating dose of bendiocarb for Ae. aegypti is changed from its initial 0.1% to 0.2% [45]. Thus, future efficacy studies of bendiocarb should target the 0.2% concentration”, have been added in the revised manuscript.

References

- I did not check correctness of each reference cited, but I noticed at least two incorrect references, therefore authors should recheck all references cited.

Response: All the references cited in the text have been checked to their correctness in the reference list.

Reviewer #3: 

The authors have a good reason for conducting this research, the manuscript is well organized and written clearly enough to be accessible to non-specialists. However, lacks a detailed description of the method used to collect larvae an

---

## [Decision Letter · Decision Letter 1]

22 May 2024

PONE-D-24-03915R1Susceptibility status of Aedes aegypti (Diptera: Culicidae) to public health insecticides in Southern Afar region, EthiopiaPLOS ONE

Dear Dr. Seid,

Thank you for submitting your manuscript to PLOS ONE. After careful consideration, we feel that it has merit but does not fully meet PLOS ONE’s publication criteria as it currently stands. Therefore, we invite you to submit a revised version of the manuscript that addresses the points raised during the review process. In particular, Reviewer 1 still felt like there was insufficient detail in describing the justification to use non-standard doses of insecticide for the susceptibility tests, and Reviewer 2 felt like there were still too many grammatical errors in the draft. Please ensure these comments are addressed in any subsequent draft versions.

We look forward to receiving your revised manuscript.

Kind regards,

James Colborn

Academic Editor

PLOS ONE

Journal Requirements:

Reviewers' comments:

Reviewer's Responses to Questions

**Comments to the Author**

1. If the authors have adequately addressed your comments raised in a previous round of review and you feel that this manuscript is now acceptable for publication, you may indicate that here to bypass the “Comments to the Author” section, enter your conflict of interest statement in the “Confidential to Editor” section, and submit your "Accept" recommendation.

Reviewer #1: (No Response)

Reviewer #2: All comments have been addressed

2. Is the manuscript technically sound, and do the data support the conclusions?

Reviewer #1: Yes

Reviewer #2: Yes

3. Has the statistical analysis been performed appropriately and rigorously? 

Reviewer #1: Yes

Reviewer #2: I Don't Know

4. Have the authors made all data underlying the findings in their manuscript fully available?

Reviewer #1: Yes

Reviewer #2: Yes

5. Is the manuscript presented in an intelligible fashion and written in standard English?

Reviewer #1: No

Reviewer #2: Yes

6. Review Comments to the Author

Reviewer #1: The authors have responded to numerous comments, but there is still a gap in explaining why the recommended discriminating doses for susceptibility tests of Aedes mosquitoes, as outlined in the WHO guideline cited in the methodology section, were not followed. This discrepancy must be addressed transparently in the methodology, providing genuine explanations for the deviation. The authors may suggest that the failure to adhere strictly to the WHO-recommended discriminating doses might be due to logistical constraints, such as the unavailability of insecticide-impregnated papers at the time of the trial. This practical impediment necessitated a pragmatic approach where alternative doses were explored. Therefore, the methodology section should clearly state the reasons for the deviation from the recommended doses.

Reviewer #2: I still noticed many grammatical errors and some typos, and have mentioned them in red fonts at some places in the PDF file to draw attention of the authors to carefully check the text.

7. PLOS authors have the option to publish the peer review history of their article (what does this mean?). If published, this will include your full peer review and any attached files.

Reviewer #1: **Yes: **Eba Alemayehu Simma

Reviewer #2: **Yes: **Dr Rajpal Singh Yadav

---

## [Author Response · Author response to Decision Letter 1]

28 May 2024

Comments on journal requirements

1.Please review your reference list to ensure that it is complete and correct. If you have cited papers that have been retracted, please include the rationale for doing so in the manuscript text, or remove these references and replace them with relevant current references. Any changes to the reference list should be mentioned in the rebuttal letter that accompanies your revised manuscript. If you need to cite a retracted article, indicate the article’s retracted status in the References list and also include a citation and full reference for the retraction notice.

Response: Dear Editor, thank you very much once again for your relevant suggestion. We have reviewed the references cited in the text to their correctness in the references list. We have also corrected a few typos of existed references particularly the volume and issue numbers in the revised manuscript. Retracted references have not been used. We have cited the WHO, 2022 of reference number[45] on future recommendation for efficacy studies of bendiocarb that should target to the 0.2% concentration. 

2. While revising your submission, please upload your figure files to the Preflight Analysis and Conversion Engine (PACE) digital diagnostic tool. 

Response: We have uploaded the figures to PACE to ensure that the figures meet PLOS requirements.

Reviewer #1 

The authors have responded to numerous comments, but there is still a gap in explaining why the recommended discriminating doses for susceptibility tests of Aedes mosquitoes, as outlined in the WHO guideline cited in the methodology section, were not followed. This discrepancy must be addressed transparently in the methodology, providing genuine explanations for the deviation. The authors may suggest that the failure to adhere strictly to the WHO-recommended discriminating doses might be due to logistical constraints, such as the unavailability of insecticide-impregnated papers at the time of the trial. This practical impediment necessitated a pragmatic approach where alternative doses were explored. Therefore, the methodology section should clearly state the reasons for the deviation from the recommended doses.

Response: Dear, Reviewer, thank you very much again for your relevant suggestion. The statements “However, we did not follow the World Health Organization recommended discriminating doses for Aedes species due to unavailability of insecticide-impregnated papers at the time of the experiment. This practical obstacle required a strategy that looked into the alternative doses for the insecticides” have been added in the revised manuscript on page 6, Line 14-18. 

Reviewer #2

 I still noticed many grammatical errors and some typos, and have mentioned them in red fonts at some places in the PDF file to draw attention of the authors to carefully check the text.

1. P1, L18 Typos correction to Non-blood fed 

Response: Dear Reviewer, thank you very much again for your important indication on grammatical and typing errors. Thus, “Non-blood fed” has been changed to “Non-blood-fed” in the revised manuscript.

2. P1, L19 “pyrethroids”, “carbamates” 

Response: “pyrethroids”, “carbamates” have been corrected to “pyrethroid” and carbamate in the revised manuscript.

3. P1, L25 “were” and L26, “were”

Response: were, were, have been corrected to “was”, “was” in the revised manuscript, because the statement explained about a single species i.e. Ae. aegypti.

4. Residual P3, L19-20

Response: “residual” has been corrected to “residual insecticide surface” in the revised manuscript.

5. P 4, L2, “base line” 

Response: “base line” has been corrected to “baseline” in the revised manuscript.

6. P7, delete the statement “The knockdown effect of pyrethroids against the Ae. aegypti were recorded during the 60 minutes exposure at an interval of 10 minutes from the start.” 

Response: “The knockdown effect of pyrethroids against the Ae. aegypti were recorded during the 60 minutes exposure at an interval of 10 minutes from the start” has been removed because of redundancy with the phenotypic susceptibility test section. 

7. P8, L3, KDT50 95% [CI] and KTD95 95%, include the 95% inside the close bracket 

Response: KDT50 95% [CI] and KDT95 95% [CI] have been corrected to KDT50 [95%CI] and KDT95 [95%CI] in the revised manuscript.

8. P8, L8, “Correct (figs 2-4) to Fig.” 

Response: (figs 2-4) has been corrected to (Figs 2-4). Only the first letter should be capitalized and no dot was needed as PLOS ONE Journal guideline for in text citation of multiple figures. 

9. P9, L4, “resistance” 

Response: “resistance” has been corrected to “to be resistant” in the revised manuscript.

10. P11, L7, “World health organization” 

Response: “World health organization” has been corrected to “World Health Organization” in the revised manuscript. 

11. P11, L10, “Add reference”: 

Response: The reference WHO, 2022 has been added in the revised manuscript.

12. P11, L11, “Mortally”

Response: “mortally” has been corrected to “mortality” in the revised manuscript.

13. P Add “%” to the insecticides in all figures 2-4 

Response: “%” has been added to the insecticides in all figures of 2-4 in the revised manuscript.

---

## [Decision Letter · Decision Letter 2]

12 Jun 2024

PONE-D-24-03915R2Susceptibility status of Aedes aegypti (Diptera: Culicidae) to public health insecticides in Southern Afar region, EthiopiaPLOS ONE

Dear Dr. Seid,

Thank you for submitting your manuscript to PLOS ONE. After careful consideration, we feel that it has merit but does not fully meet PLOS ONE’s publication criteria as it currently stands. Therefore, we invite you to submit a revised version of the manuscript that addresses the points raised during the review process. Although the reviewers felt that their major comments had been addressed, there were still many grammatical errors that need to be addressed prior to publication. The reviewers recommended having a native english speaker review for content. Please ensure the grammar and language are of publication quality in the final submitted version.

We look forward to receiving your revised manuscript.

Kind regards,

James Colborn

Academic Editor

PLOS ONE

Journal Requirements:

Reviewers' comments:

Reviewer's Responses to Questions

**Comments to the Author**

1. If the authors have adequately addressed your comments raised in a previous round of review and you feel that this manuscript is now acceptable for publication, you may indicate that here to bypass the “Comments to the Author” section, enter your conflict of interest statement in the “Confidential to Editor” section, and submit your "Accept" recommendation.

Reviewer #1: All comments have been addressed

Reviewer #2: All comments have been addressed

2. Is the manuscript technically sound, and do the data support the conclusions?

Reviewer #1: Yes

Reviewer #2: Yes

3. Has the statistical analysis been performed appropriately and rigorously? 

Reviewer #1: Yes

Reviewer #2: I Don't Know

4. Have the authors made all data underlying the findings in their manuscript fully available?

Reviewer #1: Yes

Reviewer #2: Yes

5. Is the manuscript presented in an intelligible fashion and written in standard English?

Reviewer #1: Yes

Reviewer #2: No

6. Review Comments to the Author

Reviewer #1: The authors have addressed my concerns, and I appreciate their thorough response. They have justified their use of the discrimination doses of insecticides according to WHO recommendations for testing insecticide susceptibility in Aedes mosquitoes. This indicates that they employed the recommended doses for testing Anopheles mosquitoes as outlined in the standard operating procedure for testing insecticide susceptibility in adult mosquitoes using WHO tube tests (WHO, 2022), rather than following the guidelines for monitoring and managing insecticide resistance in Aedes mosquito populations: Interim Guidance for Entomologists (WHO, 2016). Therefore, I suggest updating the reference accordingly.

Reviewer #2: I have inserted numerous corrections in the PDF file. Authors should carefully look at the suggested corrections to further improvise the manuscript.

Additionally, two specific comments are as follows:

1. At the end of many lines, words are broken e.g. on page 1 lines 13, 14, 21, 24. Please correct these errors in the text throughout.

2. Page 10: line 6: Aedes aegypti is generally endophilic and endophilic whereas the authors have stated that they are generally exophilic and exophagic. Can they support this statement with any published information?

7. PLOS authors have the option to publish the peer review history of their article (what does this mean?). If published, this will include your full peer review and any attached files.

Reviewer #1: **Yes: **Eba Alemayehu Simma

Reviewer #2: **Yes: **Dr Rajpal Singh Yadav

---

## [Author Response · Author response to Decision Letter 2]

3 Jul 2024

Comments on journal requirements

 Response: Dear Editor, We warmly thank You for Your valuable efforts to improve our manuscript to meet the PLOS ONE’s journal standard. 

We have reviewed the references cited in the text to their correctness in the reference list. We have included the reference WHO 2022, number [34] on the Phenotypic susceptibility status of Aedes aegypti section of the revised manuscript as suggested by Reviewer #1. Retracted articles have not been used in the manuscript. We have also carefully checked the grammar and typos in the revised manuscript. We believed that the manuscript meet PLOS ONE’s publication standard. 

Reviewer #1 

 The authors have addressed my concerns, and I appreciate their thorough response. They have justified their use of the discrimination doses of insecticides according to WHO recommendations for testing insecticide susceptibility in Aedes mosquitoes. This indicates that they employed the recommended doses for testing Anopheles mosquitoes as outlined in the standard operating procedure for testing insecticide susceptibility in adult mosquitoes using WHO tube tests (WHO, 2022), rather than following the guidelines for monitoring and managing insecticide resistance in Aedes mosquito populations: Interim Guidance for Entomologists (WHO, 2016). Therefore, I suggest updating the reference accordingly.

Response: Dear Reviewer, We would like to thank you for your time and efforts you made to enhance the quality of our manuscript in these series of revisions. 

We have updated the reference WHO 2022 [34] on the method section in the revised manuscript as suggested. 

Reviewer #2

I have inserted numerous corrections in the PDF file. Authors should carefully look at the suggested corrections to further improvise the manuscript. Additionally, two specific comments are as follows:

1. At the end of many lines, words are broken e.g. on page 1 lines 13, 14, 21, 24. Please correct these errors in the text throughout.

Response: Dear Reviewer, We highly appreciated you for your time and efforts that you made to enhance the quality of our manuscript in these series of revisions. 

We have corrected the broken words throughout the revised manuscript as suggested. We have checked the grammar and correct typos in the revised manuscript. 

2. Page 10: line 6: Aedes aegypti is generally endophilic and endophilic whereas the authors have stated that they are generally exophilic and exophagic. Can they support this statement with any published information?

Response: Dear, Reviewer thank you again for the question. 

We have re-phrased the statement, “Aedes aegypti is generally considered as exophilic and exophagic and daytime biters” as “Aedes aegypti is considered by some authors to be highly exophilic, exophagic, and a daytime biter”. The references number [38] and [39] cited in this study supported the high exophilic and exophagic-ness of Ae. aegypti. For instance, the authors in the reference 38 identified that 95.4% of blood-fed females of Ae. aegypti formosus subspecies were collected from outdoor sites whereas, only 4.6% were from indoor sites. In addition, 84.5% of the blood-fed females Ae. aegypti aegypti were collected from outdoor sites and 14.5 from the indoors. Thus, this suggested that the more exophilic/exophagic-ness of Aedes aegypti as a species.

---

## [Decision Letter · Decision Letter 3]

12 Aug 2024

Susceptibility status of Aedes aegypti (Diptera: Culicidae) to public health insecticides in Southern Afar Region, Ethiopia

PONE-D-24-03915R3

Dear Dr. Seid,

We’re pleased to inform you that your manuscript has been judged scientifically suitable for publication and will be formally accepted for publication once it meets all outstanding technical requirements.

Kind regards,

James Colborn

Academic Editor

PLOS ONE

Additional Editor Comments (optional):

Reviewers' comments:

Reviewer's Responses to Questions

**Comments to the Author**

1. If the authors have adequately addressed your comments raised in a previous round of review and you feel that this manuscript is now acceptable for publication, you may indicate that here to bypass the “Comments to the Author” section, enter your conflict of interest statement in the “Confidential to Editor” section, and submit your "Accept" recommendation.

Reviewer #1: All comments have been addressed

2. Is the manuscript technically sound, and do the data support the conclusions?

Reviewer #1: Yes

3. Has the statistical analysis been performed appropriately and rigorously? 

Reviewer #1: Yes

4. Have the authors made all data underlying the findings in their manuscript fully available?

Reviewer #1: Yes

5. Is the manuscript presented in an intelligible fashion and written in standard English?

Reviewer #1: Yes

6. Review Comments to the Author

Reviewer #1: The authors have addressed the comments in the revised version of the manuscript. It is in now in a good form and ready for acceptance for publication.

7. PLOS authors have the option to publish the peer review history of their article (what does this mean?). If published, this will include your full peer review and any attached files.

Reviewer #1: No

---

## [Editor Report · Acceptance letter]

15 Aug 2024

PONE-D-24-03915R3 

PLOS ONE

Dear Dr. Seid, 

I'm pleased to inform you that your manuscript has been deemed suitable for publication in PLOS ONE. Congratulations! Your manuscript is now being handed over to our production team.

Kind regards, 

on behalf of

Dr. James Colborn 

Academic Editor

PLOS ONE